# Meta-analysis reveals glucocorticoid levels reflect variation in metabolic rate, not 'stress'

Blanca Jimeno[1,2]*, Simon Verhulst[3]

[1]Instituto de Investigación en Recursos Cinegéticos (IREC), CSIC-UCLM-JCC, Ciudad Real, Spain; [2]Instituto Pirenaico de Ecologia (IPE), CSIC, Avda. Nuestra Señora de la Victoria, Jaca, Spain; [3]University of Groningen, Groningen, Netherlands

**Abstract** Glucocorticoid (GC) variation has long been thought to reflect variation in organismal 'stress,' but associations between GCs and Darwinian fitness components are diverse in magnitude, direction, and highly context-dependent. This paradox reveals our poor understanding of the causes of GC variation, contrasting with the detailed knowledge of the functional consequences of GC variation. Amongst an array of effects in many physiological systems, GCs orchestrate energy availability to anticipate and recover from predictable and unpredictable environmental fluctuations and challenges. Although this is mechanistically well-known, the extent to which GC levels are quantitatively explained by energy metabolism is unresolved. We investigated this association through meta-analysis, selecting studies of endotherms in which (1) an experiment was performed that affected metabolic rate and (2) metabolic rate and GC levels were measured simultaneously. We found that an increase in metabolic rate was associated with an increase in GC levels in 20 out of 21 studies (32 out of 35 effect sizes). More importantly, there was a strong positive correlation between the increases in metabolic rate and GCs (p=0.003). This pattern was similar in birds and mammals, and independent of the nature of the experimental treatment. We conclude that metabolic rate is a major driver of GC variation within individuals. Stressors often affect metabolic rate, leading us to question whether GC levels provide information on 'stress' beyond the stressor's effect on metabolic rate.

*For correspondence:
bjimenorev@gmail.com

**Competing interest:** The authors declare that no competing interests exist.

## eLife assessment

This work presents a **fundamental** meta-analysis on the causes of glucocorticoid variations in birds and mammals. It provides **convincing** evidence that an increase in metabolic rates increases glucocorticoid concentrations. The work will be of broad interest to animal physiologists.

## Introduction

### The accepted – but often overlooked – metabolic role of glucocorticoids

Glucocorticoid hormones (GCs; e.g. cortisol, corticosterone) were identified by Hans Selye (1907–1982) as the key molecular mediators of the 'stress reaction' and named in reference to their capacity to increase glucose in blood. Selye highlighted the fundamental role of GCs in the 'general adaptation syndrome,' that is, 'the physiological mechanisms that help to raise resistance to damage as such, irrespective of the specific nature of the damaging agents' (*Selye, 1950*). In this view, GCs play a role in 'adapting' to the challenge by triggering mechanisms that help the organism return to, or maintain, homeostasis after an environmental challenge. Indeed, towards the later stages of his career, Selye's

definition of stress was 'the nonspecific response of the body on any demand on it' (*Selye, 1976*), making the point that very different stimuli (i.e. 'stressors') triggered the same response. However, in the literature this perspective has changed over time, and 'stress' and, by extension, GCs have been widely linked to negative outcomes (but see *Koolhaas et al., 2011*; *Herman, 2022*; *MacDougall-Shackleton et al., 2019*). Consequently, researchers in fields from biomedicine to conservation physiology and animal husbandry have focused on GCs to find proxies of 'physiological and/or psychological stress' to evaluate physical and/or welfare status. Indeed, GCs have predominated over other traits throughout the stress physiology literature, to the point of being referred to as vertebrate 'stress hormones' (*Madliger et al., 2015*; *Madliger and Love, 2014*; *McCormick and Romero, 2017*). While several authors have argued against this simplified view of GC regulation during the past decades (e.g. *Koolhaas et al., 2011*; *MacDougall-Shackleton et al., 2019*; *Herman, 2022*; *Romero, 2004*; *Landys et al., 2006*; *Bonier et al., 2009*), such association prevails, in the sense that publications still abound in which GC levels are assumed to provide information on organismal stress.

The focus on GCs to measure organismal stress can be understood from their role as key mediators of organismal responses to challenges, triggering a cascade of effects on many physiological systems (*Koolhaas et al., 2011*; *Sapolsky et al., 2000*; *Deviche et al., 2017*; *Zimmer et al., 2019*; *Zimmer et al., 2020*). Furthermore, acute increases in circulating GCs are a defining aspect of the so-called 'stress response' (*Koolhaas et al., 2011*; *Sapolsky et al., 2000*). Consequently, higher GC levels circulating in plasma or deposited in keratinized tissues (i.e. hair or feathers) have traditionally been interpreted as an indication of homeostatic unbalance, poor condition, implicating low fitness prospects (reviewed in *Schoenle et al., 2021*; *Romero and Fairhurst, 2016*). However, the latter assumption is at best poorly supported by the literature, with associations between GCs and fitness (survival/reproduction) being diverse in direction and often nonexistent (*Schoenle et al., 2021*; *Zimmer et al., 2022*; *Busch and Hayward, 2009*; *Bonier et al., 2009*; *Petrullo et al., 2022*). This inconsistency raises the question what alternative inference can be made from GC variation. The urgency of this question is further underlined by the observation that GCs also increase in response to experiences we would not normally qualify as stress; for example, sexual activity induces a GC increase in humans, horses, and rodents (*Siciliani, 2000*; *Colborn et al., 1991*; *Buwalda et al., 2012*). We suggest that this question can potentially be resolved through a better understanding of physiological mechanisms and environmental factors driving GC variation.

*Selye, 1976* emphasized that homeostatic challenges of any kind lead to increases in body demands, loosely defined as 'the rate at which we live at any one moment.' Activation of the hypothalamus–pituitary–adrenal (HPA) axis represents a primary hormonal response to homeostatic challenges that, through the release of GCs, mobilizes the resources needed to fuel the current or anticipated rise in energy expenditure (current and anticipatory responses) or recover from an immediate threat that induced an unanticipated increase in energy expenditure (reactive response) (*Herman et al., 2016*; *McEwen and Wingfield, 2003*). In this context, GCs are involved in the metabolism of most types of energy reserves, modulating glucose, fat, and protein metabolism in liver, skeletal muscle, and other target tissues (*Box 1*). Allusions to the energetic role of GCs and their tight link to energy expenditure are common in physiological and ecological studies, especially those using approaches which underline the adaptive function of GC responses (*McEwen and Wingfield, 2003*; *Landys et al., 2006*; *Romero et al., 2009*; *Deviche et al., 2017*), but the extent to which GC variation can be quantitatively explained as facilitator of variation in energy expenditure has rarely been addressed (but see *Jimeno et al., 2018*; *Malkoc et al., 2021* supplementary information). We here schematically review the role of GCs in energy metabolism (*Box 1*) and investigate this link quantitatively through meta-analysis.

## Revisiting the associations between GCs and metabolic rate: A meta-analytic approach

Although previous evidence supports the link between energy expenditure and GC secretion (e.g. *Koolhaas et al., 2011*; *Sapolsky et al., 2000*; *Beerling et al., 2011*; *Buwalda et al., 2012*; *Jimeno et al., 2018*; *Malkoc et al., 2021* supplementary information), the qualitative importance of both this association and the underlying processes to explain GC variation remains unexplored; this is, however, fundamental towards accurately interpreting GC variation. We here test whether changes in energetic demands are associated with variation in GC levels. Specifically, we (i) use a meta-analytic approach to test whether experimental manipulations leading to increases in metabolic rate (MR) in

## Box 1. Glucocorticoids (GCs) and energy metabolism

We here consider GC regulation from the perspective of their role in fuelling metabolic rate. When metabolic rate is low, for example, during periods of inactivity, circulating GCs are maintained at low levels and glucose from fuel stores is released in the blood stream at a low rate matching the modest metabolic needs (*permissive* actions; *Sapolsky et al., 2000*; *Box 1—figure 1*). An increase in metabolic rate can be anticipated or unanticipated (GCs will exert *preparative* or *stimulating* actions, respectively; *Sapolsky et al., 2000*; *Box 1—figure 1*), and acute or gradual. Unanticipated but gradual increases in metabolic rate will occur, for example, when thermoregulatory costs unexpectedly increase.

In both gradual and acute increases in metabolic rate, a main role of GCs is to increase circulating glucose at a rate matching the metabolic requirements through diverse mechanisms. Decreasing plasma glucose levels trigger a series of hormonal changes that promote a switch in energy usage. Together with a decrease of insulin (in mammals, less so in birds), GCs are released into the circulation (*Andrews and Walker, 1999*; *Rosmond and Björntorp, 2000*) reducing anabolic insulin actions (*Vegiopoulos and Herzig, 2007*). Blood glucose level then increases, both by mobilization from existing stores and by inhibition of further storage. GCs also inhibit glucose uptake and glycogen synthesis in the liver, redirecting resources to gluconeogenesis and glycogenolysis, along with glucagon and catecholamines as part of the most immediate acute response. Catecholamines act quickly and increase within seconds to induce the release of energy needed to fuel the response (*Herman et al., 2016*; *Romero and Beattie, 2022*; *Sapolsky et al., 2000*). The GC response lags in time – as GCs are produced de novo at the adrenal and take minutes to be secreted – and lasts substantially longer, depending on active (feedback signaling) and passive (GC degradation) processes (*Herman et al., 2016*; *Box 1—figure 1*), enhancing and prolonging the increase in blood glucose (*Nonogaki, 2000*; *Romero and Beattie, 2022*), or recovering energy stores after a brief burst of activity. In addition, inhibition of peripheral glucose transport and utilization in response to GCs increases the availability for other tissues, such as the brain (reviewed in *Sapolsky et al., 2000*; *Herman et al., 2016*). GCs also act in other substrates, further increasing lipolysis by inducing hormone-sensitive lipase (*Slavin et al., 1994*), and reducing lipoprotein lipase activity in peripheral fat depots. They also promote pre-adipocyte differentiation, pro-lipogenic pathway activity, and cellular hypertrophy in central fat (*Vegiopoulos and Herzig, 2007*), as well as decreased thermogenesis in brown adipose tissue (*Soumano et al., 2000*). In various muscle types, GCs suppress protein synthesis while promoting protein degradation and amino acid export. When the energetic and substrate requirements of the organism are further increased (e.g. during fasting or illness), muscle tissue (40% of total body mass) becomes a rich source of amino acids, which can be mobilized as substrates for energy generation, gluconeogenesis, and protein synthesis (*Kuo et al., 2013*).

Given the existing evidence on the metabolic role of GCs, and the different time scales associated with the kind of response (anticipatory vs. perceived) as well as the hormones and substrates' physiological actions (see above), GC concentrations cannot be expected to always reflect the 'immediate' energy expenditure. However, we would expect changes in energetic demands to always require a GC response/input to meet the derived metabolic needs (*Box 1—figure 1*). This prediction of a strong association between GCs and metabolic rate, however, is not well researched and does not necessarily imply that one trait affects the other per se, as their interplay is likely to be shaped by the environmental or physiological context. Note further that we make no distinction between baseline and stress-induced GC levels, and thereby in effect assume these to be points in a continuum from a metabolic perspective; a perspective supported by the monotonic effects of GCs on glucose uptake and fat depletion (*Kattwinkel and Munck, 1966*; *Dallman et al., 1993*). Additionally, although we consider GCs to be regulated to meet energetic demands, we are aware that GCs have

many complex downstream effects at both baseline and stress-induced levels, besides energy mobilization (*Box 1—figure 1*).

**Box 1—figure 1.** Schematic representation of the association between metabolic rate and plasma levels of glucocorticoids and glucose.
Green arrows represent increasing effects, whereas red arrows represent reducing effects.

endotherms also lead to an increase in GCs (qualitative approach). We included only experimentally induced increases of energy expenditure to avoid potential masking effects of anticipatory responses or delayed effects of GCs. Because MR and GCs can fluctuate rapidly, we targeted MR and GC measurements taken simultaneously or when animals could be assumed to be in the same physiological state (e.g. within the same day and experimental treatment). We further investigated (ii) whether the magnitude of the experimentally induced changes in MR and GCs was correlated (quantitative approach) through meta-regression. Our predictions are that (i) increases in MRs are associated with increases in plasma GC concentrations, (ii) changes in GCs are proportional to induced changes in MR, and (iii) the association between increases in MR and GCs is independent of the treatment used to increase the MR.

## Results

Among the studies selected for inclusion in the analysis, the treatment effect size on MR was on average 1.85 ± 0.87 (*Figure 1—figure supplement 1*). In accordance with prediction, effects on GCs were positive in the majority of cases (32/35; *Figure 1*), and consequently the overall average effect size deviated significantly from zero, with the average GC effect size estimated at 0.73 ± 0.11 (*Table 1*). There was a strong association between MR effect sizes and GC effect sizes (*Table 1*, *Figure 2*), thus confirming prediction (ii). It is further worth noting that the residual heterogeneity did not exceed the level expected by chance (*Table 1*). MR Cohen's D was ln-transformed (see 'Materials and methods') to normalize the distribution (*Figure 2*), and AICc of models including ln MR were significantly lower compared to models including untransformed MR Cohen's D (AICc = 70.58 vs. 74.97, respectively).

The association between MR and GC effect sizes remained statistically significant when adding taxa, before/after, experiment/control effect, metabolic variable, or treatment type one by one to the model. Furthermore, none of these variables had a significant effect on GC effect size, nor did the association between MR and GC effect sizes depend on those factors (i.e. interactions between these variables and MR effect sizes were always nonsignificant; *Table 2*, *Supplementary file 4*, *Figure 2—figure supplements 1 and 2*). The latter result confirms prediction (iii). Given that none of these effects significantly improved the model, the final model when removing all factors was the one including MR effect size as only predictor of GC effect size (*Table 1*). Despite these modulators being nonsignificant, the associations were in the expected directions, with studies including within-individual variation (i.e.

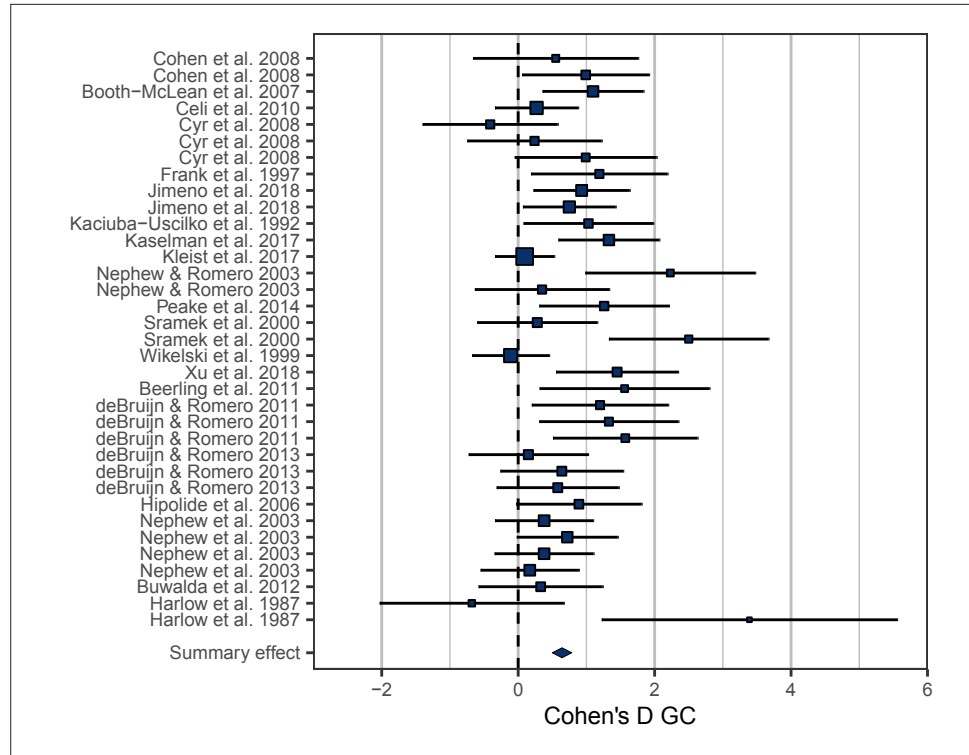

**Figure 1.** Forest plot showing the glucocorticoid (GC) effect sizes (Cohen's D ± 95% CI) associated with experimental manipulations of metabolic rate, grouped by treatment group and study. Area of squares is proportional to the experiment sample size (1/s.e.).

The online version of this article includes the following figure supplement(s) for figure 1:

**Figure supplement 1.** Forest plot showing the metabolic rate (MR) effect sizes (Cohen's D ± 95% CI) associated with experimental manipulations of MR, grouped by treatment group and study.

experiment/control effect), and not including a before/after effect reporting higher GC effect sizes (table in *Supplementary file 4*, *Figure 2—figure supplement 2*).

## Discussion

Finding a consistent functional interpretation of GC variation has proven challenging, and to this end we presented a simplified framework focusing on the interplay between energy metabolism and GCs (*Box 1*). Based on this framework, we made three predictions that we tested through a meta-analysis of studies in endotherms in which MR was manipulated and GCs were measured at the same time. The analysis confirmed our predictions, showing that experimental manipulations that increased MR induced a proportional increase in GCs (*Figure 2*), and our interpretation of this effect is that GC secretion facilitated increases in MR. This association indicates that fluctuations in energy turnover are a key factor driving variation in GC levels. From this perspective, the many downstream effects of GCs

**Table 1.** Meta regression model testing the association between metabolic rate (MR) effect sizes and glucocorticoid effect sizes.

|  | Estimate | s.e. | Z | p | 95% CI |
|---|---|---|---|---|---|
| Intercept | 0.72 | 0.11 | 6.40 | <0.0001 | 0.50–0.95 |
| MR effect size (ln) | 0.31 | 0.10 | 2.94 | 0.003 | 0.10–0.51 |

Variance components: Study.ID (Sigma^2)– Estimate = 0.00, sqrt = 0.00, n = 21.
Residual heterogeneity: QE(df = 33) = 28.40, p=0.70.
Test of moderators: QM(df = 1) = 8.64, p=0.003.

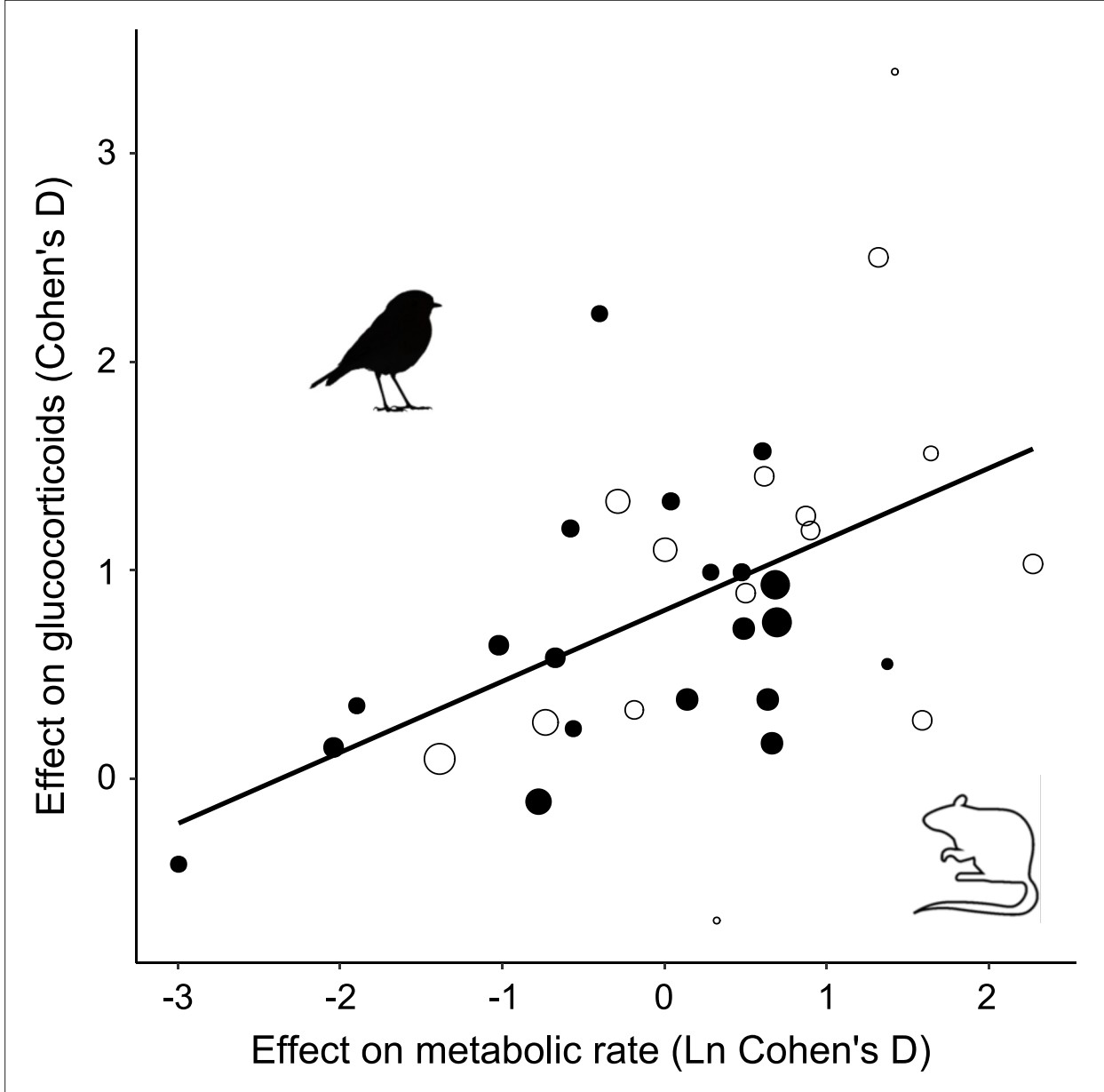

**Figure 2.** Glucocorticoid (GC) effect size (Cohen´s D) increases with increasing metabolic rate effect size similarly in studies of mammals (open circles) and birds (closed circles). Area of dots is proportional to the experiment sample size (i.e. square root of the number of individuals in which GCs were measured).

The online version of this article includes the following figure supplement(s) for figure 2:

**Figure supplement 1.** Relationship between metabolic rate and glucocorticoid effect sizes (Cohen's D) across studies.

**Figure supplement 2.** Relationship between metabolic rate and glucocorticoid effect sizes (Cohen's D) across studies as a function of before/after effect (left panel; open circles and dashed line for studies including a time effect; closed circles and continuous line for studies not including a time effect; see 'Materials and methods') and experiment/control effect (right panel; open circles and dashed line for studies including within-individual variation; closed circles and continuous line for studies not including within-individual variation).

(e.g. downregulation of immune function and reproduction; *McEwen and Wingfield, 2003*; *Sapolsky et al., 2000*) may be interpreted as allocation adjustments to the metabolic level at which organisms operate. Specifically, within-individual blood GC variation signals the MR at which the organism is functioning to all systems in the body. In this light, downstream effects of GCs can be interpreted as evolved responses to MR fluctuations, reallocating resources in the face of shifting demands on the whole organism level.

**Table 2.** Table showing the main effects of all variables considered (metabolic rate [MR], taxa, time effect, within-individual variation, metabolic variable, and treatment type) to modulate glucocorticoid effect sizes across studies. Full models are shown in *Supplementary file 4*.

| Variable | p |
|---|---|
| MR effect size (ln) | 0.003 |
| Taxa | 0.63 |
| Before/after | 0.49 |
| Experiment/control | 0.68 |
| Metabolic variable | 0.94 |
| Treatment type | |
| Treat. 2 | 0.93 |
| Treat. 3 | 0.60 |

The effect of MR on GC levels was independent of the type of manipulation used to increase MR, confirming our third prediction. Note, however, that confirmation of this prediction relied on the absence of a significant effect, and absence of evidence is not evidence of absence. However, the residual heterogeneity of our final model did not deviate from a level expected due to sampling variance, providing additional support for our third prediction.

Note that in the context of our analysis we made no distinction between 'baseline' and 'stress-induced GC levels (*Box 1*). Firstly, because these concepts are not operationally well defined – baseline GC levels are usually no better defined than 'not stress-induced.' Secondly, when considering the facilitation of MR as a primary driver of GC regulation, there does not appear a need to invoke different classes of GC levels instead of the more parsimonious treatment as continuum. This is not to say that this also applies to the functional consequences of GC-level variation: it is well known that receptor types differ in sensitivity to GCs (*Landys et al., 2006*; *Sapolsky et al., 2000*; *Romero, 2004*), thereby potentially generating step functions in the response to an increase in GC levels.

We restricted the meta-analysis to experimental studies and expect the association between MR and GCs to be less evident in a more natural context. Associations between GCs and MR will be most evident when animals are maintained at different but stable levels of MR because then the rate at which tissues are fuelled is likely to be in equilibrium with the metabolic needs. While equilibrium conditions can be created in laboratory studies, conditions will usually be more variable in the wild. When MR fluctuates, for example, due to short-term variation in activity, GC variation will track MR fluctuations, but with a time lag (*Box 1*), thus adding complexity to the MR/GC association and its detectability. Furthermore, experiments yield estimates of associations within the average individual in the study, while data collected in a natural context usually rely on variation between individuals (but see *Malkoc et al., 2022*). Associations between individuals will be less strong than the associations within individuals due to individual variation in GC levels and GC reactivity (e.g. *Liu et al., 1997*; *Weaver et al., 2004*; *Yehuda et al., 2014*; *Taff et al., 2018*; *Taff et al., 2022*). Between-subject variation is likely to be even larger on an interspecific level, and in line with this expectation the comparative evidence on the MR/GC association is mixed. While a strong positive MR/GC correlation was reported for mammals (*Haase et al., 2016*), *Francis et al., 2018* found no consistent GC/MR association in birds and other tetrapod taxa. However, recent comparative analyses showed avian GC variation to be positively correlated with estimated thermoregulatory costs (*Rubalcaba and Jimeno, 2022a*), and GC variation in lizards to be positively correlated with body temperature, which directly influences MR in ectotherms (*Rubalcaba and Jimeno, 2022b*). The contrast between the findings may be due to the MR and GC data not always being collected on individuals in a comparable state. We emphasize therefore the importance of measuring MR and GCs when animals are in the same state, preferably by measuring both variables at the same time.

GCs increased in the studies included in our meta-analysis in response to an induced increase in MR, but GCs can also increase in response to an *anticipated* increase in MR (*Box 1*). For example, the early morning GC increase in humans, known as the 'cortisol awakening response' (*Fries et al., 2009*), can be interpreted as preparation for an increase in MR – indeed, human 'early birds' show higher GC levels than 'night owls' in the hour after awakening (*Kudielka et al., 2006*). Likewise, GC levels increase in athletes preceding competition (*van Paridon et al., 2017*), although separating effects of psychological stress from anticipated metabolic needs is difficult in this context. Experiments in which animals are trained to anticipate an increase in MR to investigate whether this generates an anticipatory increase in GCs would be an interesting additional test of the framework laid out in *Box 1*.

Given that GC levels are often assumed to provide information on organismal 'stress' and welfare, the question arises whether the observed pattern can be the consequence of effects on psychological stress instead of MR. This question arises because manipulations of energy expenditure are always 'indirect,' in that an external treatment is used to induce an increase in MR, as opposed to a direct manipulation of MR, and this leaves room for other factors to cause the observed effects. While we acknowledge that it is not possible to demonstrate conclusively that a process is not happening, we consider it unlikely that 'stress effects' explain our findings. Firstly, because the way MR was manipulated varied widely between studies but manipulation type had no discernible effect on the MR/GC association. This was also the conclusion of one of the studies included in our meta-analysis, designed specifically to compare MR/GC associations between two MR increasing treatments, ambient temperature and noise as psychological stressor (*Jimeno et al., 2018*). Secondly, the finding that the GC increase was proportional to the increase in MR can only be explained by psychological stress when the induced psychological stress was proportional to the induced MR. Thirdly, the pattern is consistent with what is known of the functional consequences of GC variation in relation to metabolic needs (*Box 1*). Lastly, diverse noninjurious psychological stressors increase MR in humans (*Sawai et al., 2007*; *Balanos et al., 2010*; *Carroll et al., 2009*), mammals (*Harris et al., 2006*; mild and unpredictable chronic stress; *García-Díaz et al., 2007*), and birds (*Jimeno et al., 2018*), explaining why GCs generally increase in response to a stressor. We conclude, therefore, that while a causal link between MR and GCs is not the only possible explanation of our findings, we argue it to be the most parsimonious explanation. Direct manipulations of MR could confirm or reject this explanation and may, for example, be achieved using thyroid hormones, which have been shown to affect MR (*Moreno et al., 2002*; *Kim, 2008*).

We selected studies in which experimental treatments affected MR, leading us to conclude that the most parsimonious explanation of our finding is that GC levels were causally related to MR. Suppose, however, that instead we reported a correlation between MR and GCs using, for example, unmanipulated individuals. The question would then be justified whether changes in GCs affected MR or vice versa. Direct effects of GCs could be studied using pharmacological manipulations. However, while many studies show that GC administration induces a cascade of effects, when the function of GCs is to *facilitate* a level of MR, as opposed to *regulate* variation in MR, we do not anticipate such manipulations to induce an increase in MR (*Box 1*). On the other hand, when MR is experimentally increased in conjunction with pharmacological manipulations that *suppress* the expected GC-increase (an experiment that to our best knowledge has not yet been done), we would predict that the increase in MR can be maintained less well compared to the same MR treatment in the absence of the pharmaceutical manipulation. This result we would interpret to demonstrate that maintaining a particular level of MR may be dependent on GCs as facilitator, but it would be misleading to interpret this pattern to indicate that GCs regulate MR, as is sometimes proposed. Additionally, it would be informative to investigate whether energy turnover immediately before blood sampling is a predictor of GC levels as we would predict on the basis of the interpretation of our findings. Increasing the use of devices and techniques that monitor energy expenditure or its proxies (e.g. accelerometers) may be a way to increase our understanding of the generality of the GC–MR association.

Authors who assumed GC levels to be a proxy of physiological stress have struggled with the interpretation of findings such as the mixed results with respect to fitness consequences of GC variation. Our findings offer a way to interpret such variation: GCs are regulated with respect to their role in facilitating energy metabolism, and we encourage researchers to approach and interpret findings from this perspective. For example, a positive association between GCs and reproductive success may indicate that individuals who are able to sustain high MRs attain higher fitness (e.g. *Bauch et al., 2016*), while a negative association indicates the opposite effect (e.g. *Ouyang et al., 2013*; see *Atema et al., 2022*, for a more general discussion of this specific contrast). Given that GCs have many other downstream effects (*Sapolsky et al., 2000*), for example, suppressing immune function (*Cain and Cidlowski, 2017*), and growth (*Allen, 1996*), this may seem an overly simplistic approach. However, in this framework, downstream effects of GCs may be understood as responses to a system-level readout of the current level of energy metabolism, with high levels affecting the allocation of energy to different energy demanding processes. In this view, the link between GCs and energy-demanding processes is asymmetric, in the sense that GCs affect energy allocation to, for example, growth, but there is no direct feedback from growth to GC levels. In conclusion, whereas GCs are widely seen as 'stress hormones,' we offer a different interpretation and question whether GC variation reveals any physiological stress beyond fluctuations in energy expenditure.

## Materials and methods

### Literature search

We reviewed the literature to identify empirical studies reporting measurements of both MR and plasma GCs. We compiled studies that met all following criteria: (1) including an experimental manipulation of any kind leading to increases in MR which was quantified (i.e. both significant and nonsignificant increases). Among these, we also included those studies reporting heart rate as a metabolic measure as heart rate and MR are strongly correlated (*Bevan et al., 1994*; *Bevan et al., 1995*; *Butler et al., 2004*; *Word et al., 2022*). (2) Including measurements of natural GC concentrations in plasma (i.e. not exogenous or chemically induced; e.g. with ACTH or CRH). (3) measurements of GCs and metabolism had to be on the same individuals and measured in the same physiological state. The latter condition excludes, for example, studies with daily energy expenditure measurements combined with GCs measured at one time point. Finally, we only included studies on endotherms (birds and mammals) because metabolic regulation differs strongly between endotherms and ectotherms.

We conducted a database search (Web of Science, July 20, 2021) to identify candidate studies using the following two combinations of search terms: "*energy expenditure*" AND (*glucocorticoid OR cortisol OR corticosterone*) and "*metabolic rate*" AND (*glucocorticoid OR cortisol OR corticosterone*). After the search, we consecutively selected articles after (1) abstract review, (2) full-text review, and (c) data availability for effect size calculations. Using this approach, we identified a total of 14 studies that met all our criteria (see *Supplementary file 1* for additional information on the number of studies obtained on each of the search steps). We also systematically checked the reference list of these 14 papers, which yielded an additional 7 papers. Thus, we included a total of 21 papers (*Supplementary file 2*) in our analyses, of which 12 were on birds and 9 on mammals. Also, 9 of the 22 papers included more than one experimental treatment, yielding a total of 35 effect sizes. For each of these studies, we extracted information on study species or metabolic and GC variables reported, among others (*Supplementary file 2*). Additionally, we recorded variation related to the experimental design, the variables that were quantified, and the type of treatment used: (1) before/after design: whether the experimental manipulation included a time effect (i.e. individuals served as their own control, being measured before and after the experimental manipulation); (2) experiment/control design: whether the experiment accounted for within-individual variation (i.e. all individuals went through all experimental treatments); (3) whether MR or heart rate was the metabolic variable; and (4) the type of treatment that induced an increase in MR (see below) (*Supplementary file 2*).

### Effect size calculations

To estimate effect sizes of metabolism and GCs, we used the web-based effect size calculator Practical Meta-Analysis Effect Size Calculator, following *Lipsey and Wilson, 2001* and *Nakagawa and Cuthill, 2007*. We calculated standardized mean-difference effect sizes (Cohen's D), which we computed from means and standard deviations (19 studies) or *t*-test (2 studies). When metrics were presented graphically only, we extracted data from the figure(s) using the GetData Graph Digitizer software (http://getdata-graph-digitizer.com/). See *Supplementary file 3* for details on data extraction and effect size calculations.

For each study, we compared the mean MR and level of plasma GCs of individuals in the treatment group(s) to that of individuals in the control group. For studies in which treatment was confounded with time, because pretreatment measurements were used as control and compared with measurements during treatment, the pretreatment measure was used as control when calculating effect sizes in studies where there was a single treatment. When studies with a before-after design included more than one experimental treatment, the treatment yielding the lowest metabolism was taken as control for the effect size calculations. Thus, confounding time with treatment was avoided whenever possible.

### Statistical analyses

We conducted all meta-analyses using the *rma.mv* function from the *metafor* package (*Viechtbauer, 2010*), implemented in R (version 4.0.1, *R Development Core Team, 2020*). Standard errors were used for the weigh factor. All models contained a random intercept for study identity to account for inclusion of multiple experimental treatments or groups from the same study. Most species were used in a single study, and we therefore did not include species as a random effect in addition to study identity. The number of species was, however, insufficient to reliably estimate phylogenetic effects; we, therefore, limited the analysis in this respect with a comparison between birds and mammals (see

below). The dependent variable was either the MR or the GC effect size. One model was fitted with the MR effect size as a dependent variable to estimate the average effect on MR across all studies in the analyses. All other models had the GC effect size as dependent variable and MR effect size as a moderator. Distribution of MR effect sizes was skewed, which was resolved by ln-transformation, which yielded a better fit compared to a model using the linear term (evaluated using AIC, see 'Results' for details). Our first GC model contained only the MR effect size as a fixed independent variable. This model provides a qualitative test of whether GC levels increase when MR is increased and tests prediction (i) by providing an estimate of the intercept, which represents the average GC effect size because we mean centered the ln-transformed MR effect size (*Schielzeth, 2010*). The same model tests prediction (ii) whether the GC effect increases with an increasing MR effect size, which will be expressed in a significant regression coefficient of the MR effect size.

Following the model with which we tested our main predictions, we ran additional models to test for the effects on GC effect sizes of (1) taxa (birds vs. mammals), two design effects, namely (2) before/after effect, and (3) experiment/control effect, (4) metabolic variable measured (MR or heart rate), and (5) type of treatment (categorized in climate, psychological, or other). This last factor tests our prediction (iii). We included these variables as modulators in the analysis, as well as the two-way interactions of these factors with the MR effect size. All factors were coded as 1 (bird/no before-after effect/no experiment-control effect/MR) or 2 (mammal/before-after effect/experiment-control effect/ heart rate), and then mean-centered. Treatment type was categorized as 1 (climate), 2 (psychological), or 3 (others). We compared models with vs. without these additional variables using Akaike's information criterion, with correction for small sample sizes (AICc, *Akaike, 1974*) for which a change in AICc of 2 is considered significant (*Burnham et al., 2011*). Models within ΔAICc < 2 were considered best-fitting models, and we further explored the effects of the main predictors when present in these top models. To rule out publication bias effects (i.e. regression test for funnel plot asymmetry), we included a weighing variable (square root of the sample size) as moderator in the models as Egger's test is not a reliable test of funnel plot asymmetry in multilevel models. Variable effects and results remained quantitatively very similar and qualitatively unchanged. Furthermore, because many labs contributed multiple studies, we tested the effect of including Lab (N = 16) as random factor in the models. However, this variable had a negligible effect on the models, and we therefore excluded it from the final models.

## Acknowledgements

BJ was funded by grant FJC2019-039748-I of MCIN/AEI/10.130 39/501100011033.

## Additional information

### Funding

| Funder | Grant reference number | Author |
|---|---|---|
| Ministerio de Ciencia e Innovación | JC2019-039748-I | Blanca Jimeno |

The funders had no role in study design, data collection and interpretation, or the decision to submit the work for publication.

### Author contributions

Blanca Jimeno, Conceptualization, Data curation, Formal analysis, Investigation, Visualization, Methodology, Writing – original draft; Simon Verhulst, Conceptualization, Formal analysis, Supervision, Funding acquisition, Investigation, Visualization, Methodology, Writing - review and editing

### Author ORCIDs

Blanca Jimeno ⬚ https://orcid.org/0000-0003-3040-0163
Simon Verhulst ⬚ http://orcid.org/0000-0002-1143-6868

Reviewer #1 (Public Review): https://doi.org/10.7554/eLife.88205.3.sa1

Reviewer #2 (Public Review): https://doi.org/10.7554/eLife.88205.3.sa2
Author Response https://doi.org/10.7554/eLife.88205.3.sa3

## Additional files

### Supplementary files

• Supplementary file 1. Study selection steps and number of studies found.

• Supplementary file 2. Table showing the information extracted from each study included in the meta-analysis. Studies included in the meta analysis: *Cohen et al., 2008*; *Booth-McLean et al., 2007*; *Celi et al., 2010*; *Cyr et al., 2008*; *Frank et al., 1997*; *Jimeno et al., 2018*; *Kaciuba-Uscilko et al., 1992*; *Keselman et al., 2017*; *Kleist et al., 2017*; *Nephew and Romero, 2003*; *Peake et al., 2014*; *Srámek et al., 2000*; *Wikelski et al., 1999*; *Xu et al., 2018*; *Beerling et al., 2011*; *de Bruijn and Romero, 2011*; *de Bruijn and Romero, 2013*; *Hipólide et al., 2006*; *Nephew et al., 2003*; *Buwalda et al., 2012*; *Harlow et al., 1987*.

• Supplementary file 3. Effect size calculations. The document Includes one sheet per study with the data extracted, the part of the article it was extracted from, and the effect size calculations and results further included in the meta analysis and *Supplementary file 2*.

• Supplementary file 4. Meta-regression model (quantitative approach) testing the effect of (a) taxa, (b) before/after effect, (c) experiment/control effect, (d) use of metabolic rate (MR) or heart rate as metabolic variable, and (e) treatment type, on the association between MR and glucocorticoid effect sizes across studies.

• MDAR checklist

### Data availability

Data generated or analyzed during this study are provided along with the manuscript, as supplementary files (Supplementary File 2 and Supplementary File 3).

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
