## [Editor Report · eLife assessment]

This work presents a **fundamental** meta-analysis on the causes of glucocorticoid variations in birds and mammals. It provides **convincing** evidence that an increase in metabolic rates increases glucocorticoid concentrations. The work will be of broad interest to animal physiologists.

---

## [Referee Report · Reviewer #1 (Public Review)]

Authors performed a meta-analysis of GC concentrations and metabolic rates in birds and mammals. They found close associations for all studies showing a positive association between these two traits. As GCs have been viewed with close links to "stress," authors suggest that this overlooks the importance of metabolism and perhaps GC variation does not relate to "stress" per se but an increase in metabolism instead.

This is an important meta-analysis, as most researchers acknowledge the link between GCs and metabolism, metabolism is often overlooked in studies. The field of conservation physiology is especially focused on GCs being a "stress" hormone, which overlooks the importance of GCs in mediating energy balance, i.e., an animal that has high GC concentrations may not be doing that poorly compared to an animal with low GC concentrations, it might just be expending more energy, e.g., caring for young. The results, with overwhelming directionality and strong effect sizes, support the link for a positive association with these two variables.

My main concern lies in that most of the studies come from a few labs, therefore there may be limited data to test this relationship. I would include lab as a random effect to see how strong this effect might be. Furthermore, I would like to see a test of the directionality of the two variables. Authors suggest that changes in metabolism affect GC levels but likely changes in GC levels would affect metabolism. Why not look into studies that have altered GC levels experimentally and see the effect on metabolism? Based on the close link, authors suggest that GCs may not play a role outside of "stress" beyond the stressor's effect on metabolic rate. However, if they were to investigate manipulations of GCs on metabolic rate, the link may or may not be there, which would be interesting to look at. I firmly believe that GCs are tightly linked to metabolism; however, I also think that GCs have a range of effects outside of metabolism as well, depending on the course and strength of the stressor.

This work helps in the thinking that GCs are not the same as a "stress" hormone or labelling hormones with only one function. As hormones are naturally pleiotropic, the view of any one hormone being X is overly simplistic.

---

## [Referee Report · Reviewer #2 (Public Review)]

Where this study is interesting is that the authors do a meta-analysis of studies in which metabolic rate was experimentally manipulated and both this rate and glucocorticoid levels were simultaneously measured. Unsurprisingly, there are relatively few such studies and many are from a single lab. More studies are needed. While the results of the analysis are compelling, they are not surprising. That said, this work is important.

---

## [Author Response]

The following is the authors’ response to the original reviews.

**Reviewer #1 (Public Review):**
The authors performed a meta-analysis of GC concentrations and metabolic rates in birds and mammals. They found close associations for all studies showing a positive association between these two traits. As GCs have been viewed with close links to "stress," authors suggest that this overlooks the importance of metabolism and perhaps GC variation does not relate to "stress" per se but an increase in metabolism instead.This is an important meta-analysis, as most researchers acknowledge the link between GCs and metabolism, metabolism is often overlooked in studies. The field of conservation physiology is especially focused on GCs being a "stress" hormone, which overlooks the importance of GCs in mediating energy balance, i.e., an animal that has high GC concentrations may not be doing that poorly compared to an animal with low GC concentrations, it might just be expending more energy, e.g., caring for young. The results, with overwhelming directionality and strong effect sizes, support the link for a positive association with these two variables.My main concern lies in that most of the studies come from a few labs, therefore there may be limited data to test this relationship. I would include lab as a random effect to see how strong this effect might be.

We think this is a good point, and we ran the main models included in the manuscript including Lab as random effect (N = 35 experiments, 21 studies, 16 labs). This did not affect the results, leading to negligible changes in the model parameters (alternative model tables are shown in Author response table 1 and 2). In the revised version of the manuscript we mention that we tested the effect of Lab but did not keep this variable in the models (lines 183-185)

**Author response table 1. sa3table1:** Meta regression model testing the association between metabolic rate (MR) effect sizes and glucocorticoid effect sizes.

	Estimate	s.e.	z	P	95% C.I..
Intercept	0.76	0.11	7.20	< 0.0001	0.55-0.96
MR effect size (In)	0.32	0.10	2.97	0.003	0.11-0.52
Variance components: outer factor: Lab(N=16), inner factor: Study.ID (N=21). tau^2- Estimate = 0.02, sqrt = 0.1					
Residual heterogeneity: QE(df=33)=28.26,p=0.70					
Test of moderators: QM(df=1)=8.83,p=0.003					

**Author response table 2. sa3table2:** Meta regression model (quantitative approach) testing the effect of (a) Taxa, (b) Before / after effect, (c) Experiment / control effect, (d) Use of Metabolic Rate or Heart Rate as metabolic variable and (e) Treatment type, on the association between metabolic rate (MR) and glucocorticoid effect sizes across studies.

(a)	Estimate	s.e.	z	P	95% C.I.
Intercept	0.72	0.11	6.30	< 0.0001	0.50-0.95
MR effect size (In)	0.30	0.11	2.75	0.006	0.09-0.52
Taxa (mammal)	0.07	0.23	0.33	0.74	-0.38-0.54
MR : Taxa	0.16	0.22	0.71	0.48	-0.28-0.59
Variance components: outer factor: Lab (N=16), inner factor: Study. ID (N=21). tau^2- Estimate = 0.00, sqrt = 0.09					
Residual heterogeneity: QE(df=31)=27.47,p=0.65					
Test of moderators: QM(df=3)=9.53,p=0.023					
(b)	Estimate	s.e.	z	P	95% C.I.
Intercept	0.77	0.10	7.69	< 0.0001	0.57-0.96
MR effect size (ln)	0.32	0.11	3.00	0.002	0.11-0.54
Before / after effect (yes)	-0.30	0.38	-0.79	0.43	-1.06-0.45
MR : Time effect	-0.00	0.31	0.05	0.96	-0.59-0.62
Variance components: Lab (N=16), inner factor: Study.ID (N=21). tau^2 - Estimate = 0.03, sqrt = 0.16					
Residual heterogeneity: QE(df=31)=27.72,p=0.64					
Test of moderators: QM(df=3)=9.39,p=0.024					
(c)	Estimate	s.e.	z	P	95% C.I.
Intercept	0.76	0.10	7.51	< 0.0001	0.56-0.96
MR effect size (ln)	0.32	0.11	2.97	0.003	0.11-0.52
Exp./ control effect (yes)	0.08	0.31	0.25	0.80	-0.53-0.69
MR : Exp. / control effect	-0.05	0.32	-0.00	0.99	-0.62-0.62
Variance components: Lab (N=16), inner factor: Study.ID (N=21). tau^2- Estimate = 0.02, sqrt = 0.15					
Residual heterogeneity: QE(df=31)=28.18,p=0.61					
Test of moderators: QM(df=3)=8.88,p=0.030					
(d)	Estimate	s.e.		P	95% C.I.
Intercept	0.76	0.11	7.12	< 0.0001	0.55-0.97
MR effect size (ln)	0.34	0.11	3.00	0.003	0.12-0.56
Met. variable (HR)	0.15	0.22	0.68	0.49	-0.28-0.59
MR : Met. variable	0.01	0.23	0.06	0.96	-0.43-0.46
Variance components: outer factor: Lab (N=16), inner factor: Study.ID (N=21). tau^2- Estimate = 0.03, sqrt = 0.16					
Residual heterogeneity: QE(df=31)=28.03,p=0.62					
Test of moderators: QM(df=3)=9.21,p=0.027					
e)	Estimate	s.e.	z	P	95% C.I.
Intercept	0.82	0.20	4.09	< 0.0001	0.43-1.21
MR effect size (In)	0.26	0.20	1.30	0.19	-0.13-0.64
Treat. Type 2	0.01	0.29	0.02	0.98	-0.57-0.58
Treat. Type 3	-0.20	0.32	-0.62	0.53	-0.81-0.42
MR: Treat. Type 2	0.09	0.27	0.34	0.74	-0.44-0.62
MR: Treat. Type 3	0.08	0.29	0.28	0.78	-0.49-0.66
Variance components: Lab (N=16), inner factor: Study.ID (N=21). tau^2- Estimate = 0.04, sqrt = 0.20					
Residual heterogeneity: QE(df=29)=27.77,p=0.53					
Test of moderators: QM(df=5)=9.32,p=0.097					

Furthermore, I would like to see a test of the directionality of the two variables. Authors suggest that changes in metabolism affect GC levels but likely changes in GC levels would affect metabolism. Why not look into studies that have altered GC levels experimentally and see the effect on metabolism? Based on the close link, authors suggest that GCs may not play a role outside of "stress" beyond the stressor's effect on metabolic rate. However, if they were to investigate manipulations of GCs on metabolic rate, the link may or may not be there, which would be interesting to look at. I firmly believe that GCs are tightly linked to metabolism; however, I also think that GCs have a range of effects outside of metabolism as well, depending on the course and strength of the stressor.

The directionality of the two variables is indeed a question of interest – we show that changes in metabolic rate affect GCs, but does the reverse also happen? In the schematic model we propose in Box 1, we propose that the effect is uni-directional, i.e. metabolic rate affects GC-levels, but GCs have no direct effect on metabolic rate. We note that there may however be an indirect effect, in that in the absence of a GC-response to an increase in metabolic rate the organism would after some time no longer be able to fuel the metabolic rate. Because we anticipate that more readers may raise this question, we have added the following paragraph to the discussion:

“We selected studies in which experimental treatments affected MR, leading us to conclude that the most parsimonious explanation of our finding is that GC levels were causally related to MR. Suppose however that instead we reported a correlation between MR and GCs, using for example unmanipulated individuals. The question would then be justified whether changes in GCs affected MR or vice versa. Direct effects of GCs could be studied using pharmacological manipulations. However, while many studies show that GC administration induces a cascade of effects, when the function of GCs is to facilitate a level of MR, as opposed to regulate variation in MR, we do not anticipate such manipulations to induce an increase in MR (Box 1). On the other hand, when MR is experimentally increased in conjunction with pharmacological manipulations that supress the expected GC-increase (an experiment that to our best knowledge has not yet been done), we would predict that the increase in MR can be maintained less well compared to the same MR treatment in the absence of the pharmaceutical manipulation. This result, we would interpret to demonstrate that maintaining a particular level of MR may be dependent on GCs as facilitator, but it would be misleading to interpret this pattern to indicate that GCs regulate MR, as is sometimes proposed. Additionally, it would be informative to investigate whether energy turnover immediately before blood sampling is a predictor of GC levels, as we would predict on the basis of the interpretation of our findings. Increasing the use of devices and techniques that monitor energy expenditure or its proxies (e.g. accelerometers) may be a way to increase our understanding of the generality of the GC-MR association. “

We based our hypotheses and searching criteria on the assumption that GCs induce physiological processes to help the organism facilitate energetic demands. Pharmacologically induced increases in GCs would lead to physiological responses and associations that we consider not comparable to the ones reported in this work, as we base our hypotheses on natural (i.e. non pharmacologically induced) GC and MR variation. This said, with exogenous GC administration, we may expect GC cascade effects, but not necessarily an increase in MR. Here - and acknowledging that the link between GCs and metabolic rate may entail complex steps - we predict that GC administration may lead to an increase in blood glucose and may affect energy allocation at a tissue-specific level. However, such increase may have no effect on whole-organism energy expenditure, unless energy expenditure is limited by glucose availability. We however acknowledge that it would be interesting to investigate the kind of associations between MR, GCs and other physiological variables (e.g. glucose) that appear when inducing an increase in GCs, as these would broaden our understanding of the mechanistic processes underlying these associations.

We show that variation in GC levels was explained by variation in MR, independent of the stimulus that caused the increase in MR. We propose that the most parsimonious interpretation of our findings is that GC variation is an indicator of variation in MR, independent of the cause of variation in MR. We do not intend to prove causality when making predictions on the co-dependency of metabolic rate and GCs. In fact, our predictions do not imply that one trait necessarily affects the other per se, as these interplay is likely to be shaped by the environmental or physiological context (Box 1). Thus, the specific mechanisms underlying how changes in metabolic rate induce changes in GCs - or the other way around - need to be investigated. One step to tackle this in upcoming research would indeed be studying the effects of exogenous GCs on metabolic rate.

In the manuscript, we clarify that GCs have a variety of cascade effects besides metabolism (Box 1). On the basis of our results, however, we suggest that many of the downstream effects of GCs may be interpreted as allocation adjustments to the metabolic level at which organisms operate (lines 235236), but we do acknowledge that these cascade effects are complex and affects many systems besides metabolism.

This work helps in the thinking that GCs are not the same as a "stress" hormone or labelling hormones with only one function. As hormones are naturally pleiotropic, the view of any one hormone being X is overly simplistic.

We fully agree, but stress that we focus on how GCs are regulated, which may be less complex than its pleiotropic functions. Indeed, we consider that the many functions of GCs have potentially clouded the question as to how GCs are regulated.

**Reviewer #2 (Public Review):**
Where this study is interesting is that the authors do a meta-analysis of studies in which metabolic rate was experimentally manipulated and both this rate and glucocorticoid levels were simultaneously measured. Unsurprisingly, there are relatively few such studies and many are from the lab of Michael Romero. While the results of the analysis are compelling, they are not surprising. That said, this work is important.It is worth noting that in this analysis, the majority of the studies, if not all, are dealing with variation in baseline levels of glucocorticoids. That means the hormone is mostly acting metabolically at these lower levels and not as a stress response hormone as it does when levels are much higher. This difference is probably due to differences in receptors being activated. This could be discussed.

As mentioned in Box 1, within our hypothesis framework we make no distinction between baseline and stress-induced GC-levels, and thereby in effect assume these to be points in a continuum from a metabolic perspective. Our results support this view, as our sample includes baseline- and stressinduced –range GC values, and these are not distinguishable (Fig. 3). We do however recognize that we did not return to this issue in the Discussion, while the same issue may well occur to many readers familiar with the literature. We therefore added the following paragraph to the discussion:

“ Note that in the context of our analysis we made no distinction between ‘baseline’ and ‘stressinduced GC-levels (Box 1). Firstly, because these concepts are not operationally well defined – baseline GC-levels are usually no better defined than ‘not stress-induced’. Secondly, when considering the facilitation of metabolic rate as primary driver of GC regulation, there does not appear a need to invoke different classes of GC-levels instead of the more parsimonious treatment as continuum. This is not to say that this also applies to the functional consequences of GC-level variation: it is well known that receptor types differ in sensitivity to GCs (Landys et al. 2006; Sapolsky et al. 2000; Romero 2004), thereby potentially generating step functions in the response to an increase in GC-levels.”

We note further that to our best knowledge there are no standard or established thresholds that allow us to separate GC levels into “baseline” and “stress-induced”, and in any case these concentration ranges differ strongly among species and experimental set-ups (e.g. captive vs. free-living individuals). Consequently, many of the studies included in our work report what would typically be interpreted as “stress-induced” levels, and thus within the range of those reported by standardized stress protocols (e.g. levels above 20-30 ng/ml for corticosterone in bird species, Cohen et al. 2007, Jimeno et al. 2018; levels between 150-300 ng/ml in captive rats, Buwalda et al. 2012, Beerling et al. 2011; levels 2-10 times above baseline in humans, Sramek et al. 1999). We also want to note that we work with effect sizes, i.e. not GC levels, and that GC measurement units differ among studies. Mean GC values by study in the original units are shown in Table S3.

**Reviewer #1 (Recommendations For The Authors):**
L26: why is the causality in this direction? Not that I don't think that metabolic rate drives GC variation but the meta-analyses here could suggest the opposite direction as well? That GC phenotype could limit or promote metabolic activity? (In terms of the natural variation studies and not the experimental ones)

See our detailed response above, on the directionality of the association and the hypotheses underlying our searching criteria and the paragraph on this topic added to the discussion.

L27: again, I am not sure the meta-analyses can lead to this question. Although there is a tight link between GC and metabolic rate, there is still variation around that is unexplained.

See our detailed response above, on the directionality of the association and the hypotheses underlying our searching criteria and the paragraph on this topic added to the discussion.

L45: I think there is plenty of literature in the field that would say that GCs are linked to metabolism and don't define GCs as synonymous with stress. See MacDougall and others that you cite later in the paragraph: "GCs and stress are not synonymous." I think maybe shifting the strong language at the beginning might help with your argument later on.

We do not disagree, but two considerations made us retain the ‘strong language’. Firstly, while many authors mention links between GCs and metabolic rate, as we read the literature, the quantitative importance of this link to understand GC variation is underestimated in our view. Secondly, the literature is rife with articles that clearly do not consider metabolic rate variation as a driver of the GC variation they observe.

Box 1: on the diagram the link between GCs and learning is problematic as there are plenty of studies that show a negative effect on learning with GC exposure. It usually depends on the time course of GCs and learning outcomes.

We agree with the referee´s point. Learning was deleted from the diagram to avoid confusion.

The diagram also suggests that GCs in the blood decreases insulin. For Aves that are rather insulin insensitive, the evidence that GCs affect insulin concentrations are very limited, even in the poultry literature.

Indeed, and we now mention in box 1 that GC effects on insulin are primarily found in mammals, and less so in birds.

Box 1 at the end also makes a point about GCs having complex downstream effects at baseline and stressinduced levels, besides energy mobilization but the abstract seems to indicate that there are limited effects of GCs outside of metabolism. Hence why I also advocate being careful about the wording in the abstract.

The related abstract sentence has been rewritten to avoid this inconsistency (lines 17-18)

L107: "being or not significant" meaning significant or not? The wording is awkward

We reworded the sentence for clarity. We included studies reporting both significant and nonsignificant increases in metabolic rate.

L110: why not look at whether experimental increases in GCs also induce increases in metabolic rate, i.e., the directionality of the two variables. (point 2)

See our detailed response above, on the directionality of the association and the hypotheses underlying our searching criteria and the paragraph on this topic added to the discussion.

The studies, although there are ~30, are overlapping in terms of labs, i.e., a lot of them came from the same lab. Did you think to include lab as a random effect to see if there are effects of one or two labs doing work that strengthened the results?

We think this is a good point, and we ran the main models included in the manuscript including Lab as random effect (N = 35 experiments, 21 studies, 16 labs). Including Lab as random factor did not affect the results, leading to negligible changes in the model parameters. We provide tables with the model results in our previous response. In the text we now mention that we tested the effect of Lab but did not keep this variable in the models (lines 183-185)

L314: I think it depends on the time course and intensity of the stressor. I firmly believe that outside of metabolic demands, high levels of GCs chronically or the inability to mount a proper stress response is indicative of pathology or something outside of metabolism.

Whether the association between GCs and MR holds under a context of ‘chronic stress’ (i.e.understood as chronically elevated GCs) remains to be tested. We note, however, that chronically high levels of metabolic rate may potentially have pathological effects.

**Reviewer #2 (Recommendations For The Authors):**
I find the title a bit misleading. The conclusion from the study is that glucocorticoid levels can reflect metabolic rate, not that glucocorticoid levels do not indicate stress. Remember, stress can certainly affect metabolic rate.

We see the point but note that other drivers of variation in metabolic rate also increase GCs, as we show in our analysis, and hence we propose that GC variation always indicate variation metabolic rate, and only stress when stress is the cause of the increase in metabolic rate.